# NeuMa, Born to Work

## Abstract

Grounded in the evolutionary principle that resource constraints favor structural solutions for complex computation, we propose a neuro-centric framework that reframes the success of State Space Models (SSMs), e.g. Mamba, as an unconscious convergence to an incomplete model of the hippocampus—the brain's canonical circuit for sophisticated computations such as pattern separation and completion. Motivated by this, we introduce NeuroMamba (NeuMa), a novel architecture that consciously and faithfully implements the canonical hippocampal circuit, including the dentate gyrus (DG), Cornu Ammonis 3 (CA3), and Cornu Ammonis 1 (CA1), using foundational SSM blocks. Enabled by custom kernels, our design bridges the gap between biological plausibility and practical efficiency. Experiments demonstrate that NeuMa achieves superior performance and learning efficiency on synthetic benchmarks. More critically, it exhibits profound biological fidelity by spontaneously replicating the "orthogonalized state machine" dynamics of the biological hippocampus. Finally, we validate its capacity for real-world scientific discovery by developing a generative agent for piezoelectric catalysis that achieves superior performance in this complex, low-resource domain, thereby showcasing a new path for AI architecture design rooted in neuroscience.

## 1 Introduction

Biological evolution offers a profound insight in resource-limited optimization. With a sparse toolkit of carbon-based elements, nature's path to sophisticated computation was not through exotic matter, but intricate structure. This principle of function emerging from corresponding structure is a necessity of the physical world, honed over eons. It marks a conceptual departure from the prevailing paradigm in AI design, which often resembles a form of modern alchemy: a trial-and-error process that, while powerful, frequently lacks a foundation in first principles and leaves interpretability a persistent challenge. We propose a paradigm shift, from this empirical alchemy to a principled construction guided by nature's proven evolutionarily-optimized blueprints, echoing Feynman's dictum: "What I cannot create, I do not understand."

Among nature's blueprints, the brain's hippocampal circuit is a paramount example of this principle in action. Its computational prowess is fundamentally predicated upon its sophisticated circuit architecture, as shown in Figure 1. The principal information influx from the neocortex, relayed via the entorhinal cortex, initiates the canonical trisynaptic loop. This pathway begins as perforant path projections converge onto dentate gyrus (DG) granule cells, which in turn dispatch powerful mossy fiber inputs to CA3 pyramidal neurons. From there, the signal is processed within CA3's extensive recurrent collaterals and is simultaneously propagated forward to CA1 pyramidal neurons via the Schaffer collateral pathway. This entire ensemble, complemented by a direct monosynaptic entorhinal-CA1 projection, materializes as a massively recurrent and convergent computational substrate that underpins the hippocampus's dual mastery over temporal sequences (memory) (Dusek & Eichenbaum, 1997; Nieh et al., 2021) and spatial arrangements (navigation) (McNaughton et al., 1983; Ferbinteanu & Shapiro, 2003)—precisely the class of complex dependency and representation learning problems that define the frontiers of modern AI, spanning both sequence modeling and computer vision.

It is this intricate structure that directly inspires this work. We realize that modern State Space Models (SSMs) such as Mamba have implicitly recapitulated functional fragments of this circuitry. As shown in Figure 2, we reframe Mamba as an unconscious, incomplete convergence toward the hippocampal model, notably omitting a dedicated input-processing stage analogous to the dentate

gyrus and a direct output pathway from the recurrent core, analogous to CA3. Advancing this biological analogy, we introduce NeuroMamba, a conscious, circuit-level implementation that accurately models these components. By translating the first principles of a biological blueprint into a modern engineering practice, we aim to endow our model with not only superior performance and robustness but also profound interpretability and computational efficiency, forging a new path for AI design.

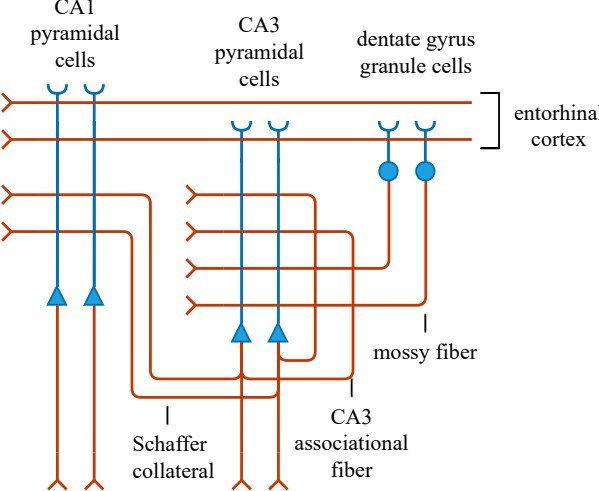

Figure 1: **The hippocampal circuit.** The principal neurons of the hippocampal formation are depicted, with circles representing granule cells and triangles representing pyramidal neurons. Blue denotes dendrites and cell bodies (postsynaptic components), while red indicates axons (presynaptic components). Synapses can be formed where red and blue lines intersect.A prominent feature is the extensive recurrent network formed by the CA3 pyramidal neurons via their axon collaterals.The perforant path, which originates from the superficial layers of the entorhinal cortex, provides the primary input to the hippocampus and converges on the CA1 region via two main pathways:(1)Monosynaptic Pathway: Axons of the perforant path synapse directly onto the dendrites of CA1 pyramidal neurons.(2)Trisynaptic Pathway: Information is serially relayed through the granule cells of the dentate gyrus and the pyramidal neurons of CA3, before reaching CA1 via the Schaffer collaterals (axonal projections from CA3).

## 2 BACKGROUND AND RELATED WORK

### 2.1 FROM QUADRATIC BOTTLENECKS TO SELECTIVE STATE SPACES

The pursuit of modeling long-range dependencies, a task at which Transformers excel (Vaswani et al., 2017) but are hindered by quadratic complexity, spurred the development of State Space Models (SSMs). The modern line of structured SSMs, from the foundational HiPPO framework (Gu et al., 2020) to S4 (Gu et al., 2022) and its variants (Gupta et al., 2022; Smith et al., 2023), achieved near-linear complexity but remained Linear Time-Invariant (LTI), with fixed dynamics unable to adapt to input content. The critical breakthrough came with Mamba (Gu & Dao, 2023), which introduced an input-dependent selection mechanism. This allows the SSM to become a time-varying system that selectively processes information, retaining linear-time scaling via a hardware-aware parallel scan and establishing an efficient foundation for sequence modeling.

### 2.2 HIPPOCAMPAL COMPUTATION AS A BLUEPRINT FOR PREDICTIVE MODELING

This work is grounded in the computational principles of the hippocampus. While foundational theories highlighted its role in pattern separation and completion (Marr & Cowan, 1991), inspiring early AI models with external memory (Graves et al., 2016), a more potent modern perspective reframes the hippocampus not as a retrospective storage device, but as a prospective, predictive model of the

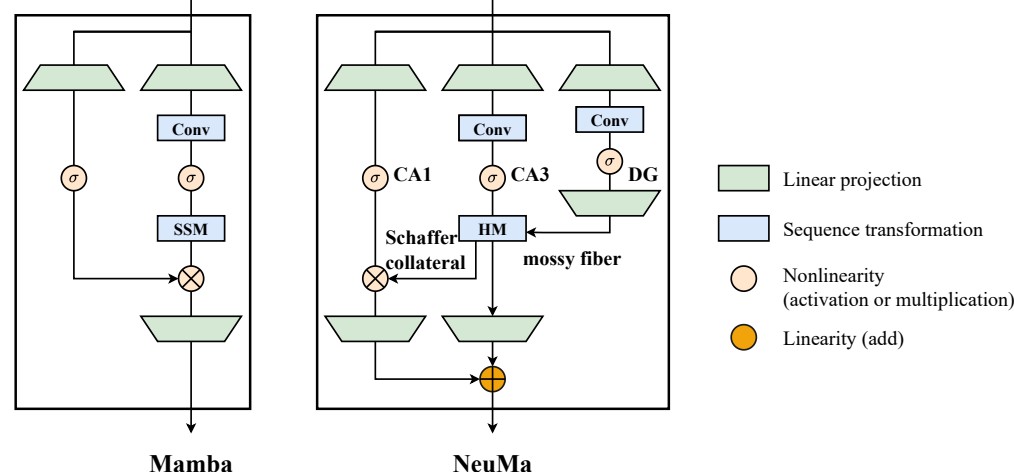

Figure 2: **Architecture.** Mamba can be viewed as a hippocampal model that omits the granule cell input branch of the dentate gyrus and a direct output from CA3. In contrast, NeuroMamba is designed to more faithfully replicate the native hippocampal circuit architecture.

world (Lisman & Redish, 2009). This view is formalized by the Successor Representation (SR) theory (Dayan, 1993; Gershman, 2018), which posits that the hippocampus learns a predictive map of future states (Stachenfeld et al., 2017). This creates a profound analogy with sequence modeling, where an SSM serves as a natural computational substrate for SR-like computations (Stoewer et al., 2022). From this perspective, Mamba's selective state updates can be seen as learning a flexible, content-dependent predictive map (Plitt & Giocomo, 2021). Our work builds directly on this analogy, showcasing that a more faithful implementation of the hippocampal circuit's architecture can yield a more powerful and interpretable predictive model.

## 3 THE NEUROMAMBA MODEL

### 3.1 OVERALL ARCHITECTURE AND INFORMATION FLOW

Departing from the monolithic structure of conventional SSMs, the NeuroMamba model herein is a novel, explicitly modular SSM architecture designed as a high-fidelity, circuit-level implementation of the mammalian hippocampal formation (Figure 2). Its components are engineered to mirror the computational roles of the DG, CA3, and CA1 fields, and its information flow faithfully mimic the hippocampus's parallel pathways: at each timestep $t$, an input signal $x_t$ (analogous to neocortical input) is projected and split into three internal streams for processing by their respective modules:

- The DG-like module receives its dedicated input stream, performs a non-linear transformation (Convolution + SiLU) aimed at pattern separation—a core computational principle of the dentate gyrus for orthogonalizing inputs (Myers & Scharfman, 2009)—and transmits its output as the mossy fiber signal $(mf)_t$, exclusively to the CA3 module.

- The CA3-like module acts as the "three-input, three-output" computational core of the circuit. It integrates three distinct input streams: a direct input from the EC, the processed mossy fiber signal from the DG, and its own recurrent state from the previous timestep. Correspondingly, it generates three outputs: the Schaffer collateral projection $(sc)_t$ to the CA1 module, the recurrent signal to itself that forms the basis of the next state, and a direct projection that contributes to the model's final output.

- The CA1-like module functions as the final integration stage. To faithfully model the non-linear dendritic integration in CA1 neurons (Poirazi et al., 2003), it employs a multiplicative gating operation to combine the Schaffer collateral signal from CA3 with the direct perforant path input. This mechanism is critical for achieving coincidence detection between these two distinct information streams (Katz et al., 2007).

**Hippocampus Microcircuit**
adapted from SSM with selective scan

Figure 3: **The Hippocampus Microcircuit (HM) Block.** The HM block implements hippocampal dynamics as a selective SSM. It integrates a direct input ($x$, perforant path) with a processed signal ($mf$, mossy fiber) to update its state ($h_t$, recurrent state of CA3). The direct input $x$ also generates the dynamic SSM parameters ($\mathbf{B}_t, \mathbf{C}_t, \mathbf{\Delta}_t$). The block features a dual-output architecture to model functional specialization: $y_{1,t}$ (Schaffer collateral) and $y_{2,t}$ (CA3 output).

In our implementation, the computational roles of the CA3 and CA1 modules are encapsulated within a central, modular unit we term the Hippocampus Microcircuit (HM) block, which serves as the recurrent core of the model as illustrated in Figure 3.

## 3.2 MATHEMATICAL FORMULATION

The HM block's dynamics (Figure 4) are governed by a selective SSM formalism. The core recurrence parameters ($\mathbf{B_t}$, $\mathbf{C_t}$ and step size $\mathbf{\Delta_t}$) are dynamically generated from the direct input stream $x_t$, allowing the model to modulate its behavior based on context. The continuous-time state matrix $\mathbf{A}$ is a learnable parameter. These are combined and discretized (by $\mathbf{\Delta_t}$) to produce $\overline{\mathbf{A}}_t$ and $\overline{\mathbf{B}}_t$.

---

**Algorithm 1** The Hippocampus Microcircuit (HM) Block

**Input:** $\mathbf{x} : (B, L, D)$, $\mathbf{mf} : (B, L, D)$
**Output:** $\mathbf{y}_1, \mathbf{y}_2 : (B, L, D)$
1: $\mathbf{A} : (D, N)$  ▷ Represents structured N × N matrix
2: $\mathbf{B} : (B, L, N) \leftarrow s_B(\mathbf{x})$
3: $\mathbf{C} : (B, L, N) \leftarrow s_C(\mathbf{x})$
4: $\mathbf{\Delta} : (B, L, D) \leftarrow \tau_\Delta(\text{Linear}_\Delta(\mathbf{x}))$
5: $\overline{\mathbf{A}}, \overline{\mathbf{B}} \leftarrow \text{discretize}(\mathbf{\Delta}, \mathbf{A}, \mathbf{B})$
6: $\mathbf{y}_1, \mathbf{y}_2 \leftarrow \text{HM}(\overline{\mathbf{A}}, \overline{\mathbf{B}}, \mathbf{C})(\mathbf{mf}, \mathbf{x})$  ▷ Time-varying: recurrence (scan) only
7: **return** $\mathbf{y}_1, \mathbf{y}_2$

---

Figure 4: **Algorithm.** The HM block dynamically generates SSM parameters from input $x$, modulates the state recurrence with $mf$, and produces the dual outputs, $y_1$ and $y_2$.

The hidden state $h_t$ (representing CA3 activity) is updated by integrating its previous state $h_{t-1}$, the direct CA3 input $x_t$, and the crucial mossy fiber signal $(mf)_t$:

$$h_t = \overline{\mathbf{A}}_t h_{t-1} + \overline{\mathbf{B}}_t x_t + (mf)_t$$

From this state, two distinct outputs are generated, modeling the circuit's functional specialization. A linear projection from $h_t$ forms the Schaffer collateral signal $(sc)_t$, which is then gated in the CA1-like module to produce $y_{\text{CA1},t}$. Concurrently, the state $h_t$ itself serves as the direct CA3-like output, $y_{\text{CA3},t}$:

$$(sc)_t = y_{1,t} = \mathbf{C}_t h_t$$
$$y_{\text{CA1},t} = (sc)_t \odot \text{SiLU}(x_{\text{CA1},t})$$
$$y_{\text{CA3},t} = y_{2,t} = h_t$$

Finally, these two pathways are projected back to the model dimension D and summed to produce the final output of the NeuroMamba layer:

$$y_t = \text{out\_ca\_one\_proj}(y_{\text{CA1},t}) + \text{out\_ca\_three\_proj}(y_{\text{CA3},t})$$

### 3.3 HARDWARE-AWARE IMPLEMENTATION

A defining feature of modern SSMs, which NeuroMamba inherits and adapts, is its hardware-aware design. The core recurrence relation of the SSM is inherently sequential, which poses a bottleneck for parallel hardware like GPUs. To overcome this, we follow the principle established by Mamba (Gu & Dao, 2023) and implement the recurrence using a highly optimized parallel scan algorithm. This operation, materialized in custom CUDA and Triton kernels, allows the model to be trained efficiently in linear time with respect to sequence length ($\mathcal{O}(L)$), akin to a convolutional network. During inference, however, it operates as an efficient recurrent model, requiring only a constant-size state and an $\mathcal{O}(1)$ update per step. This dual-mode efficiency, bridging parallelizable training and fast autoregressive generation, enables our biocircuit-inspired architecture to scale to the demands of modern machine learning tasks.

## 4 EXPERIMENTS

We conduct a series of experiments on synthetic benchmarks to validate NeuroMamba's efficiency. We first establish its superior performance and learning dynamics against baseline models, then demonstrate its unique ability to replicate biological learning phenomena, and finally use targeted ablations to provide causal evidence for the functional roles of its architectural components.

### 4.1 FOUNDATIONAL CAPABILITIES: BENCHMARKING COMPARING TO SSMS

#### 4.1.1 SELECTIVE COPYING TASK

The Selective Copying task tests a model's ability to recall informational tokens from a long distractor sequence. Full data generation and hyperparameter details are provided in Appendix B.1.

**Results and Analysis.** As shown in Figure 5b, in a parameter-constrained regime (D = 24) where baseline Mamba struggles, NeuroMamba demonstrates markedly superior learning. It converges significantly faster, achieving a 97% accuracy milestone in nearly 40% fewer steps. Furthermore, its learning dynamic is more stable, avoiding the prolonged initial performance plateau exhibited by Mamba. This suggests NeuroMamba's structured architecture provides a beneficial inductive bias, leading to more robust and efficient optimization.

#### 4.1.2 INDUCTION HEADS TASK

The Induction Heads task tests a model's ability to infer and apply abstract, non-local associative rules under a demanding length extrapolation protocol. The full multi-level task design is detailed in Appendix B.2.

**Results and Analysis.** Our experiments on the algorithmically complex Level 2 revealed a striking phenomenon (Figure 5c). NeuroMamba successfully solved the task, achieving near-perfect

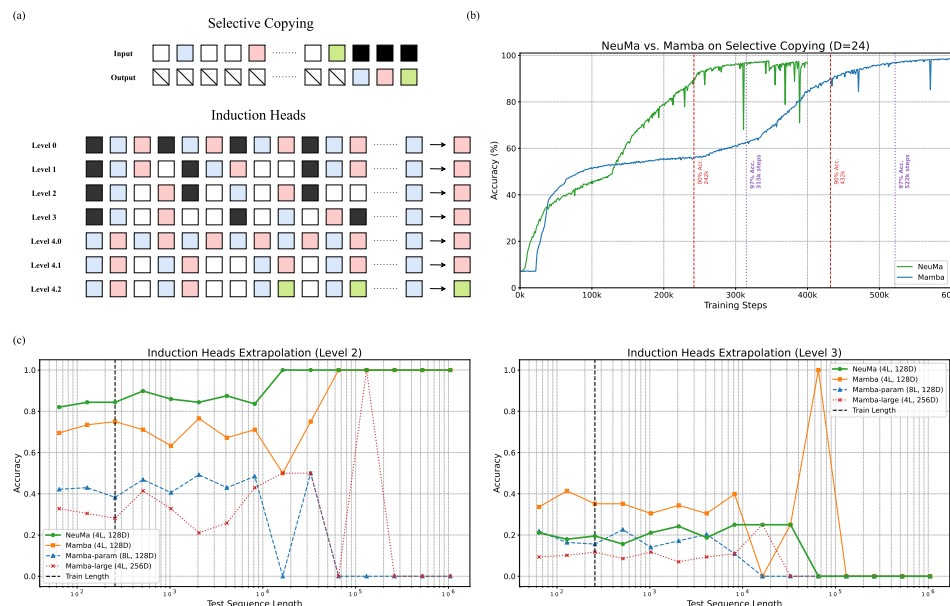

Figure 5: **Performance on Foundational Benchmarks.** (a) Schematics for the Selective Copying and Induction Heads tasks. (b) On Selective Copying, NeuMa (green) converges significantly faster and more stably than Mamba (blue), which exhibits a prolonged learning plateau. The late-stage performance spikes unique to NeuMa are reminiscent of biological coincidence detection, an emergent form of input-timing-dependent plasticity (Golding et al., 2002; Katz et al., 2007) (c) On Induction Heads Level 2, NeuMa demonstrates robust algorithmic generalization, while Mamba variants fail to converge effectively. On the harder Level 3, all models struggle. See Appendix B.2.3 for a full breakdown across all sub-tasks.

accuracy and low loss across all extrapolation lengths. In contrast, Mamba models failed to converge to an equally robust solution. While the smallest Mamba variant achieved 100% accuracy on the longest sequences, its loss remained substantially higher, indicating less confident predictions. Counter-intuitively, increasing Mamba's parameter count actively degraded its long-sequence generalization. We interpret this as evidence for the critical role of inductive bias: NeuroMamba's architectural priors provide a "scaffold" for algorithmic reasoning, whereas Mamba's less structured architecture, when given excessive capacity, is probably more prone to overfitting statistical artifacts of the short training sequences. The even more challenging Level 3, which requires suppressing long-range interference, proved insurmountable for all models, a limitation we attribute to our model representing an isolated hippocampal circuit, lacking the top-down control from cortical areas like the prefrontal cortex (PFC) (Sigurdsson & Duvarci, 2015), which is consistent with the Complementary Learning Systems (CLS) theory (McClelland et al., 1995; McClelland, 1998).

Table 1: Algorithmic isomorphism between the Induction Heads Level 2 task and the 2ACDC task.

| Computational Component | Induction Heads Level 2 | 2ACDC Task |
|---|---|---|
| **Trigger / Cue** | Special prefix token P | Visual indicator cue |
| **Key Information** | The identity of token A | The identity of the indicator |
| **Ambiguous Delay** | Interspersed noise tokens N | Featureless grey corridor walls |
| **Correct Output** | Recalling associated token B | Licking at the reward zone |
| **Core Challenge** | Learning a non-local, abstract, trigger-based conditional rule | |

### 4.1.3 SCIENTIFIC FIDELITY: REPLICATING HIPPOCAMPAL LEARNING DYNAMICS

To test if NeuroMamba's success stems from a genuine alignment with biological computation, we tasked it with replicating the core findings of a landmark neuroscience experiment on the two-alternative cue-delay-choice (2ACDC) task (Sun et al., 2025). This task, which is algorithmically isomorphic to Induction Heads Level 2 (Table 1), forces the hippocampus to form distinct, orthogonal representations for perceptually identical inputs based on prior context. The key biological finding was the emergence of an "orthogonalized state machine" in CA1 neurons, characterized by a specific temporal sequence of decorrelation. Our goal was to see if NeuroMamba could spontaneously replicate this non-trivial learning dynamic. Full task implementation and analysis protocols are in Appendix B.3.

**Results and Analysis.** The results (Figure 6) provide powerful evidence for our neuro-centric design.

- **NeuroMamba Partially Recapitulates Biological Dynamics:** Only when analyzing the CA1-like module's output could NeuroMamba consistently pass the rigorous dual-threshold (performance and decorrelation) criteria adopted from the original study. The successful runs achieved a remarkable final decorrelation (mean correlation 0.074) and, more compellingly, perfectly replicated the precise temporal sequence of learning observed in mice (Off-diagonal $\rightarrow$ Pre-R2 $\rightarrow$ Pre-R1). Fidelity to the process of learning, not just its outcome, provides a much stronger form of evidence for scientific fidelity.

- **Baseline Mamba Exhibits a Threefold Failure:** In stark contrast, Mamba failed decisively, exhibiting a threefold failure in biological fidelity: (1) it could not consistently meet the dual-threshold criteria—a characteristic akin to the inconsistent neural maps that fail to support reliable memory in biological circuits (Malone et al., 2024); (2) its learning was unstable, with final correlation values being extremely dispersed—a chaotic instability starkly different from the slow, functional representational drift observed in biological circuits (Ziv et al., 2013); and (3) it completely failed to replicate the correct temporal learning sequence.

This demonstrates that the ability to spontaneously and reliably replicate these emergent neural dynamics is a direct consequence of the specific architectural priors engineered into NeuroMamba.

## 4.2 ABLATION STUDIES: PROBING THE CIRCUIT

To provide causal evidence for the functional roles of our architectural innovations, we conducted targeted ablation studies.

- **On Selective Copying,** removing the DG pathway slightly improved performance, while removing the CA3-Out pathway impaired it (Figure 7a). This suggests the DG is a specialized module not required for simple signal filtering, while the CA3-Out is a general stabilizer.

- **On Induction Heads,** all three ablated variants, unlike the full model, were brittle and suffered catastrophic failure at specific long sequences ($2^{17}$ and $2^{19}$), as shown in Figure 7b. This reveals that the complete circuit is necessary for emergent systemic robustness.

- **On the 2ACDC task,** all ablated models decisively failed, consistently unable to pass the dual-threshold success criteria. This failure, which precluded any analysis of their learning process, stands as a key finding: the full circuit is a prerequisite for guiding the model into a stable, biologically plausible learning regime (see Appendix C.1 for visual confirmation via correlation heatmaps).

**Synthesis.** Taken together, these experiments constitute a powerful computational double dissociation. The DG pathway is a specialized module: its removal helps on a simple filtering task but is critical for robustness and biological fidelity on complex tasks. The CA3-Out pathway is a universally critical stabilizer, as its removal consistently degrades learning stability and final performance across all tested tasks. NeuroMamba's components thus form an integrated, synergistic system, where specialization and stability work in concert to create a robust and powerful architecture.

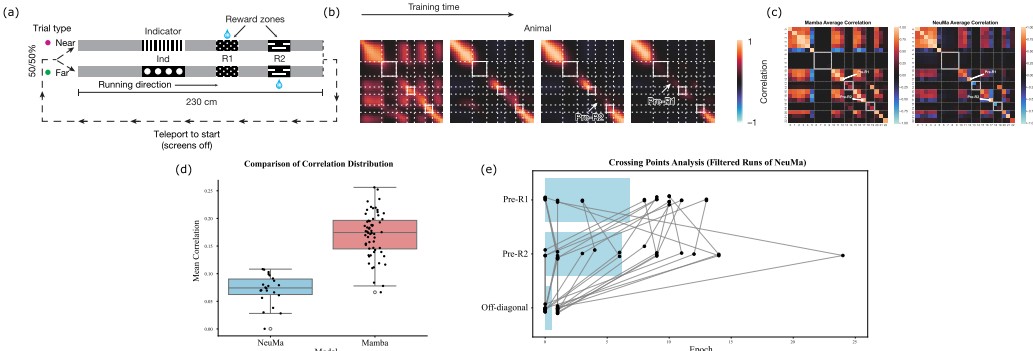

Figure 6: **The two-alternative cue-delay-choice (2ACDC) task.** (a) Schematic of the 2ACDC task, where mice must use subtle cues to resolve perceptual ambiguity on two tracks and find the correct reward (R1 or R2) (adapted from Sun et al. (2025)). (b) In the biological hippocampus, neural representations decorrelate in a stereotyped sequence: first between different gray zones (Off-diagonal), then before the far reward (Pre-R2), and finally before the near reward (Pre-R1) (adapted from Sun et al. (2025)). (c) Final mean correlation matrices for Mamba (left) and NeuMa (right), computed between the 23 timesteps of the "Near" and "Far" trials. NeuMa's darker matrix indicates stronger representation orthogonalization. (d) Box plot comparing final mean correlation across all successful runs. NeuMa achieves significantly lower correlation, confirming more complete orthogonalization than Mamba. (e) Temporal analysis shows NeuMa replicates the precise biological decorrelation sequence (Off-diagonal → Pre-R2 → Pre-R1) by tracking when key regions first cross a decorrelation threshold of 0.6. No Mamba runs met the same stringent criteria for this analysis, suggesting an architectural limitation in emulating these complex dynamics.(see Appendix B.3.4 for detailed analysis)

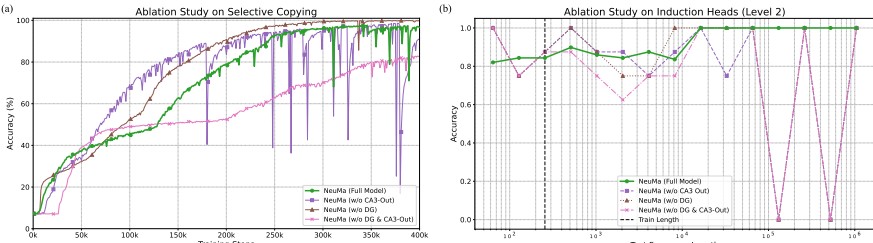

Figure 7: **Ablation studies reveal functional specialization and emergent robustness.** Performance of the full NeuMa model versus its ablated variants. **(a) On Selective Copying,** ablating the specialized DG pathway improves convergence, while ablating the CA3-Out pathway impairs learning stability. **(b) On Induction Heads,** only the full model is robust to length extrapolation; all ablated variants catastrophically fail at specific long sequences ($2^{17}$, $2^{19}$). Critically, on the **2ACDC task (data not shown),** all ablated models also failed to meet the biological fidelity criteria, demonstrating that the complete circuit is a prerequisite for robust, biologically plausible learning.

### 4.3 REAL-WORLD VALIDATION: FROM PRE-TRAINING TO SCIENTIFIC DISCOVERY

To validate the practical utility of NeuroMamba beyond synthetic benchmarks, we undertook a three-stage process to build a specialized generative agent: pre-training a foundation model, fine-tuning for conversational ability, and second-stage fine-tuning on a specialized scientific domain.

#### 4.3.1 PRE-TRAINING A FOUNDATION MODEL

We pre-trained a 140M parameter version, NeuMa-140M, to serve as a versatile foundation. The model was trained on a large-scale, multilingual, and multi-domain corpus of approximately 2.5 billion tokens. This pre-training phase was designed to endow the model with a broad base of

knowledge and reasoning capabilities, establishing a robust starting point for subsequent fine-tuning. Full details are provided in Appendix D.2.

Before fine-tuning, we benchmarked our $\approx$140M foundation model against a comparable Mamba. As shown in Table 2, NeuroMamba is substantially more efficient. This advantage stems from our structure-function principle, which yields a "superblock" so powerful that equivalent capacity is achieved with far fewer layers (12 vs. 26). This confirms our thesis that a superior local (block-level) design induces greater global (model-level) efficiency. A detailed analysis is provided in Appendix D.1.

Table 2: **Efficiency Benchmark ($\approx$140M Models).** Comparison of NeuMa and Mamba on a Nvidia 5070Ti GPU with BF16 precision. Values are reported as mean $\pm$ std. dev. over 5 runs. NeuMa's shallower, more complex layers result in superior and more consistent efficiency.

| Model | Params (M) | Layers | Training (SeqLen 2048) | | Inference (GenLen 100) | |
|---|---|---|---|---|---|---|
| | | | Throughput (tok/s) | Peak VRAM (GB) | Latency (ms/tok) | Throughput (tok/s) |
| Mamba | 136.7 | 26 | $28,380 \pm 196$ | 3.00 | $16.86 \pm 2.80$ | $1,134 \pm 199$ |
| **NeuMa (this work)** | **140.6** | **12** | **$34,236 \pm 321$ (+21%)** | **2.75 (-8%)** | **$7.16 \pm 0.18$ (>2.3x faster)** | **$2,198 \pm 182$ (+94%)** |

#### 4.3.2 FINE-TUNING FOR SCIENTIFIC DOMAIN KNOWLEDGE

To adapt our foundation model for scientific applications, we performed a "domain-acclimatization" fine-tuning step. NeuMa-140M was fine-tuned on a large corpus of scientific literature (Chem-Data700K) to create NeuMa-Chem, a new baseline imbued with foundational knowledge in chemistry and materials science. Full hyperparameter details for this stage are provided in Appendix D.3.

#### 4.3.3 AGENT-DRIVEN SCIENTIFIC DISCOVERY

NeuMa-Chem was then transformed into a generative agent for piezoelectric $CO_2$ reduction via a second fine-tuning stage on a private, multi-source dataset (see Appendix D.4 for methodological principles). The success of this approach has opened a new research avenue, the specifics of which are detailed in a forthcoming publication. We engaged this agent in an interactive loop, where human intuition guided the exploration of its generated hypotheses. This collaborative process led us to a significant breakthrough: investigating the timing of dopant introduction during the synthesis of 5% tellurium-doped barium titanate.

Our agent-guided experiments revealed that introducing the dopant during the second step of a two-step hydrothermal method dramatically enhances catalytic performance. The space-time yield for the conversion of $CO_2$ to CO increased from the literature's best of 31.6 $\mu$mol/(g $\cdot$ h) (Ma et al., 2022) to 52.34 $\mu$mol/(g $\cdot$ h), and further to 56.22 $\mu$mol/(g $\cdot$ h) with a sacrificial agent (approx. 1.7x and 1.8x the undoped rate, respectively). This result not only sets a new state-of-the-art but validates NeuroMamba as a powerful tool for accelerating real-world scientific discovery.

## 5 CONCLUSION

Our work is grounded in a fundamental lesson from biology: under the constraints of a limited elemental toolkit, the evolution of sophisticated computation is a story of structural innovation. By translating this first principle into a modern engineering practice, we introduced NeuroMamba and demonstrated that this paradigm of principled construction yields profound benefits. Our experiments confirmed that this neuro-centric design not only surpasses baselines in efficiency and algorithmic reasoning but, more critically, exhibits remarkable biological fidelity. This entire endeavor culminated in the agent-driven discovery of a novel catalytic process, achieving a state-of-the-art space-time yield of 56.22 $\mu$mol/(g $\cdot$ h). A detailed discussion is provided in Appendix A.

The limitations observed in our isolated hippocampal model point directly to the future: building more complete cortico-hippocampal circuits. This pursuit of greater biological fidelity promises not just more powerful models, but a new generation of AI that reasons with neuro-scientific plausibility, ultimately transforming our ability to understand the natural world and accelerate discovery within it.

## REPRODUCIBILITY STATEMENT

We are committed to ensuring the reproducibility of our work. The architecture of our core model, NeuroMamba, is detailed in Section 3, with pseudocode for its central computational block provided in Figure 4. To facilitate verification, we will provide the complete source code as supplementary material, which includes the implementation of the NeuroMamba model and the code for all synthetic benchmark tasks (Section 4.1) and ablation studies (Section 4.2). The detailed settings for all experiments, including data generation procedures, model training hyperparameters, and specific evaluation protocols, are thoroughly described in Appendices B, C, and D. The pre-training (Section 4.3.1) and first fine-tuning (Section 4.3.2) stages utilize public datasets, as specified in Appendix D. The final scientific discovery application (Section 4.3.3) involves a private dataset for the second fine-tuning stage; its associated code and model weights are reserved for a forthcoming publication, but the core methodological principles are discussed in Appendix D.4.

## USE OF LARGE LANGUAGE MODELS (LLMS)

A large language model (LLM) was used as a writing-assist tool during the preparation of this paper. The use of the LLM was limited to improving grammar, polishing phrasing, ensuring consistency in terminology, and reorganizing sentences to improve clarity based on the authors' intent. All core scientific ideas, theoretical frameworks, model designs, experimental protocols, results analysis, and conclusions were conceived, designed, and executed exclusively by the human authors. The LLM did not play any role in research ideation in this work.

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

# A    EXTENDED DISCUSSION AND BROADER IMPLICATIONS

Our work is built upon the same first principle that guides biological evolution: when constrained to a simple set of building blocks, the only path to sophisticated computation is through intricate structure. Applying this lens to AI, we propose a new paradigm for designing and interpreting State Space Models. We reframe Mamba as an unconscious, incomplete analogue of the hippocampus—nature's paramount example of a structural solution—and present NeuroMamba as a conscious engineering of its well-understood biological circuit.

The novelty of NeuroMamba lies in its introduction of a dual-selectivity mechanism. It retains the fine-grained, input-dependent selectivity within the core SSM block, as pioneered by Mamba, while introducing a new, higher-level structural selectivity achieved by routing information through functionally specialized streams. This shift from a monolithic to a modular, multi-pathway architecture is our core contribution. It is this principled design—not merely mimicking biology, but translating its computational strategies—that underpins the model's success.

A key feature of NeuroMamba's design is its distribution of parameters. Compared to a monolithic SSM like Mamba, which scales capacity primarily through the vertical stacking of identical, parameter-lean layers, a NeuroMamba layer is intentionally more parameter-intensive. This is not computational overhead, but a direct consequence of our structure-function design principle. These additional parameters are allocated to the functionally specialized pathways (e.g., the DG-like pattern separator, the CA3 direct output). As our ablation studies (Section 4.2) demonstrate, these supposedly "costly" components are, in fact, indispensable for the superior performance, robustness, and biological fidelity we observe. Our work thus advocates for a paradigm that embraces not just "vertical depth" but also "horizontal complexity"—building fewer, but more powerful and principled, computational circuits per layer. Our empirical benchmarks (Section 4.3.1) confirm that this philosophy of 'horizontal complexity' translates directly into superior computational efficiency, validating our design choices not only in theory but also in practice.

The synergistic roles of NeuroMamba's components, revealed through our ablation studies, provide a clear mapping from structure to function. The DG-like module, implementing the principle of pattern separation, is not merely a feature extractor but a specialized circuit whose utility is task-dependent—beneficial for algorithmic reasoning and biological fidelity, but an unnecessary overhead for simple signal filtering. The CA3-like module's core computation, which integrates past memory ($h_{t-1}$), current sensation ($x_t$), and a summarized, orthogonalized version of the current sensation ($(mf)_t$), emerges as the central engine for complex temporal reasoning. This "triple-input integration" provides a mechanistic explanation for the model's ability to handle non-local, abstract dependencies.

Furthermore, the CA1-like module's multiplicative gating operation is a direct implementation of a fundamental neural computation principle: coincidence detection. It provides a biological grounding for the gating mechanisms that have proven empirically successful in models like LSTMs, suggesting an instance of convergent evolution between engineered and biological solutions. Our work, therefore, consciously derives a mechanism that other architectures may have discovered through trial-and-error.

Beyond its architectural novelty, the true significance of our work lies in its validation as a tool for real-world scientific discovery. The journey from a foundational model to a specialized scientific agent highlights a critical dimension of our contribution: interpretability fosters trust. Because NeuroMamba's operations are grounded in understandable neuro-computational principles (e.g., pattern separation, coincidence detection), it transforms from a "black box" predictor into a comprehensible research partner. This trust is the prerequisite for scientists to engage with and build upon AI-generated hypotheses, a crucial step in accelerating the cycle of discovery. This paradigm, where neuro-inspired models serve as trustworthy, collaborative agents, holds immense potential to be generalized to other complex domains, such as drug discovery and systems biology.

Notably, the limitations of our model are as informative as its successes. The shared failure on the Induction Heads Level 3 task and the imperfect replication of the final representational state in the 2ACDC task both point to the same profound conclusion: the inherent computational boundaries of an isolated hippocampal circuit. These complex tasks, which require robust suppression of long-range interference, are believed to necessitate top-down cognitive control from the prefrontal

cortex (PFC). Thus, our model's "failures" are not merely technical shortcomings but valuable scientific findings that validate its biological plausibility and chart a clear path forward. The most exciting future direction is the development of more complete cortico-hippocampal models, where a NeuroMamba-like module is integrated with a PFC-like module that provides attentional modulation and executive control. This represents a principled step towards unlocking the next generation of sophisticated, general-purpose sequential reasoning in AI.

## B EXPERIMENTAL DETAILS

### B.1 SELECTIVE COPYING TASK

As described in Section 4.1.1, the Selective Copying task evaluates a model's ability to recall specific informational tokens from a long sequence of distractors. Our experimental setup is designed to test the models in a parameter-constrained regime, highlighting the differences in learning efficiency and dynamics.

#### B.1.1 DATA GENERATION

The synthetic data for this task was generated on-the-fly for each training and evaluation batch, following a process adapted from the original S4 implementation(Gu et al., 2022). For each sequence, we set the vocabulary size ($n_{tokens}$) to 16. The input sequence was constructed by first sampling 16 informational tokens ($l_{memorize} = 16$) from the vocabulary range [1, 15]. These tokens were then randomly inserted into a long distractor sequence composed of 4096 padding tokens ($l_{noise} = 4096$), where the padding token is 0. This configuration corresponds to the variable=True setting, ensuring informational tokens are not simply front-loaded. To signal the prediction phase, 16 marker tokens (value 15) were appended to the end of this combined sequence. The model's objective is to predict the original 16 informational tokens in their correct order. We used a fixed memorization length, so the variable_length parameter was set to False. The input tokens were fed to the model as integer IDs rather than one-hot vectors (one_hot = False).

#### B.1.2 MODEL AND TRAINING HYPERPARAMETERS

This experiment was specifically designed to compare NeuroMamba and the baseline Mamba in a challenging, low-parameter regime. The NeuroMamba model was configured with 2 layers (n_layer = 2) and a model dimension (d_model) of 18. The gate and controller expansion factor (expand_gc) was set to 2. Other architectural settings included enabling RMS normalization (rms_norm = True), using 32-bit precision for residual connections (residual_in_fp32 = True), and employing a fused add-and-norm operation (fused_add_norm = True). Input and output embeddings were not tied (tie_embeddings = False). For a fair comparison, the baseline Mamba model was configured with a similar parameter count, using a model dimension (d_model) of 24 and 2 layers. Both models were trained for a total of 400,000 steps. We used the Adam optimizer with a constant learning rate of 1e4 and a batch size of 64. The loss was calculated using the standard cross-entropy function. To monitor progress, model performance was evaluated every 1,000 training steps. Each evaluation consisted of calculating the average loss and accuracy over 100 newly generated validation batches. For complete reproducibility of our results, the global random seed was fixed to 42.

### B.2 INDUCTION HEADS TASK

As described in Section 4.1.2, the Induction Heads task is a diagnostic benchmark designed to assess a model's ability to infer and apply abstract algorithmic rules. Our implementation follows the length extrapolation protocol established in prior work, training models exclusively on short sequences and evaluating them on exponentially longer ones.

#### B.2.1 DATA GENERATION

The data generation process was designed as a comprehensive diagnostic suite with systematically increasing difficulty. For all levels, the vocabulary size was set to 16, and each sequence was constructed by creating 3 unique key-value association pairs. Noise, when introduced, consisted of random tokens sampled from the vocabulary, with the length of each noise segment varying randomly from 1 to 4 tokens. The difficulty levels are defined as follows:

- **Level 0 (Baseline):** This level tests basic, trigger-based recall. Sequences consist of clean, adjacent triplets of the form [P, A, B], where P is a special prefix token.

- **Level 1 (Memory Robustness):** To test retention against interference, random noise tokens are inserted between the [P, A, B] triplets.

- **Level 2 (Abstract Pattern Recognition):** This level tests the ability to learn a non-local association rule. Noise is inserted within each triplet, between the prefix, key, and value (e.g., [P, N, A, N, B]).

- **Level 3 (Combined Stress Test):** This is the most difficult trigger-based scenario, combining the challenges of Levels 1 and 2 by inserting noise both within and between the triplets.

- **Level 4 (Autonomous Learning Suite):** In this advanced suite, the explicit prefix token P is removed, compelling the model to discover the associative structure autonomously. This suite includes three sub-levels:

  1. **Level 4.0 (Sanity Check):** Clean [A, B] pairs to test for basic unsupervised pattern discovery.
  2. **Level 4.1 (Robust Discovery):** Noise is inserted between the [A, B] pairs, testing the model's ability to find associations in a noisy context.
  3. **Level 4.2 (Dynamic World Modeling):** This level directly tests memory updating and temporal reasoning. A key is first associated with one value and later re-associated with a new value (e.g., [A, B], ..., [A, C]). The model is then queried with A and must correctly predict C.

During training, all models were exposed exclusively to sequences of length 256. For evaluation, we generated a fixed set of validation sequences with lengths scaling exponentially from 64 ($2^6$) up to 1,048,576 ($2^{20}$). The model's task is to predict the correct value token given a query containing the corresponding key token. The loss is calculated only on this single prediction token.

### B.2.2 MODEL AND TRAINING HYPERPARAMETERS

We evaluated multiple model configurations to ensure a fair and comprehensive comparison. Our primary NeuroMamba model, which successfully solved the Level 2 task, was configured with 4 layers (n_layer = 4) and a model dimension of 128 (d_model = 128). The gate/controller expansion factor was set to 2 (expand_gc = 2). For comparison, we also trained Mamba variants with an identical architecture (4 layers, D = 128), a similar parameter count (8 layers, D = 128), and a significantly larger capacity (4 layers, D = 256). Standard architectural settings were used for all models, including RMS Normalization (rms_norm = True), 32-bit precision for residual connections (residual_in_fp32 = True), and a fused add-and-norm operation (fused_add_norm = True). The vocabulary size was padded to a multiple of 8 for efficiency (pad_vocab_size_multiple = 8), and input and output embeddings were tied (tie_embeddings = True). All models were trained for a total of 204,800 steps using the AdamW optimizer with a batch size of 8. The learning rate was set to 2e-4, and weight decay was set to 0.0. The model's performance on the fixed validation sets was logged every 8,192 training steps. The global random seed was fixed to 42 for all experiments to ensure reproducibility.

### B.2.3 GRAPHICAL ANALYSIS OF HYPERPARAMETER SENSITIVITY

To provide a more granular view of our findings on the Induction Heads task, this section presents the detailed performance plots for both Mamba and NeuroMamba across all seven sub-tasks. These figures supplement the analysis in Section 4.1.2 by visualizing the full spectrum of results. Specifically, we illustrate how Mamba's performance changes with architectural scaling (varying layers and hidden dimensions) and how NeuroMamba's performance is influenced by its key neuro-inspired hyperparameter, the DG-pathway expansion factor (`expand_gc`). This detailed comparison aims to offer a comprehensive resource for understanding the models' behaviors under different configurations.

Figure 8: **Detailed Performance on Induction Heads Sub-tasks.** This series of figures provides a comprehensive visual comparison of Mamba and NeuroMamba across all seven sub-tasks of the Induction Heads benchmark, illustrating the impact of architectural hyperparameters on performance. Each figure shows a specific sub-task, with the left column displaying Mamba's performance as its architecture scales (layers and dimension), and the right column showing NeuroMamba's performance as its DG-pathway expansion factor (expand_gc) is varied.

## Level 0 (Baseline)

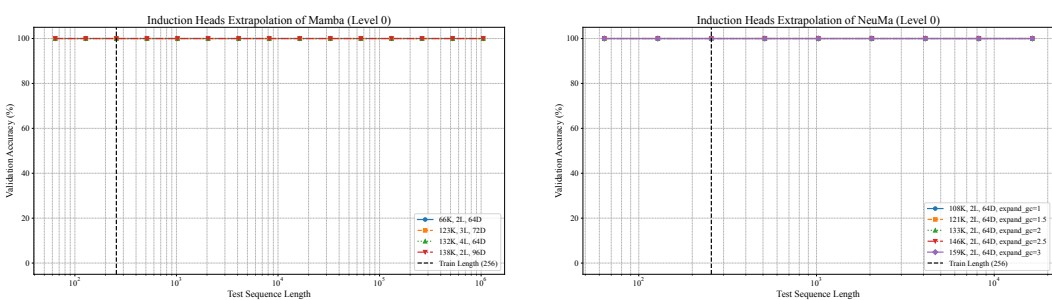

Figure 9: Baseline performance comparison between Mamba (left) and NeuroMamba (right).

## Level 1 (Memory Robustness)

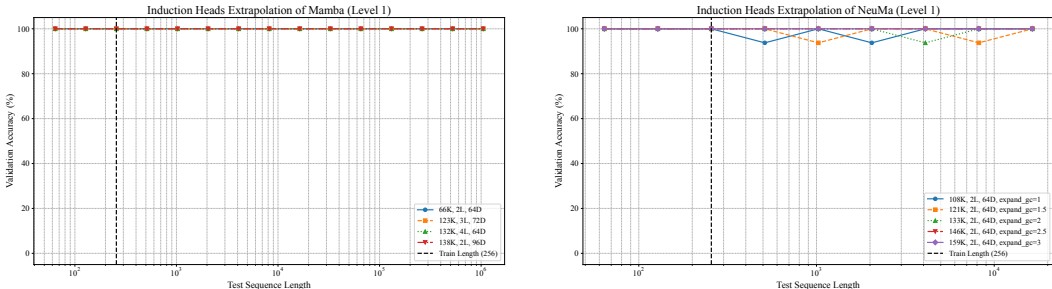

Figure 10: Memory robustness test results showing Mamba (left) and NeuroMamba (right) performance under memory-intensive conditions.

## Level 2 (Abstract Pattern Recognition)

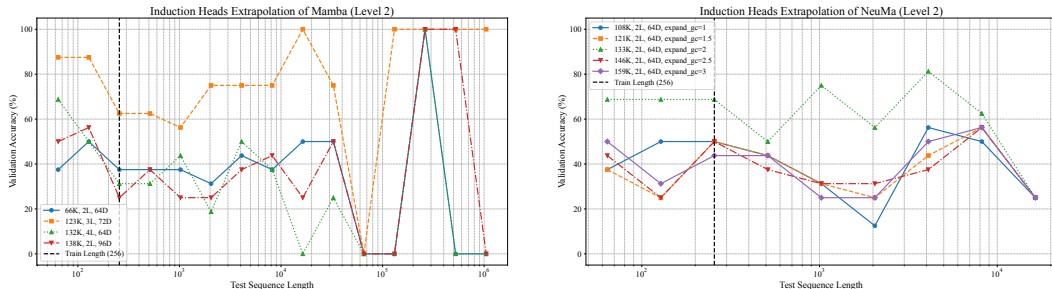

Figure 11: Abstract pattern recognition capabilities of Mamba (left) versus NeuroMamba (right).

**Level 3 (Combined Stress Test)**

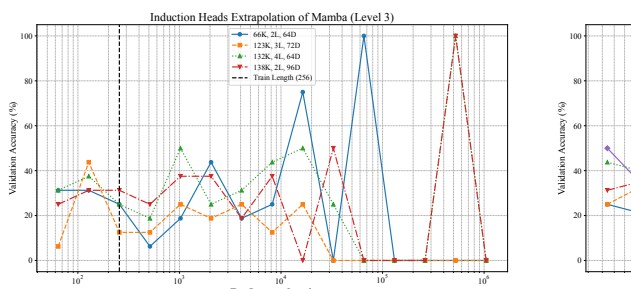 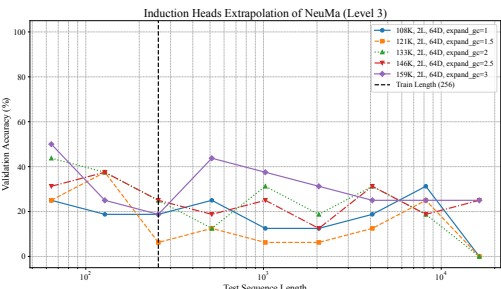

Figure 12: Combined stress test results comparing Mamba (left) and NeuroMamba (right) under multiple challenging conditions.

**Level 4.0 (Sanity Check)**

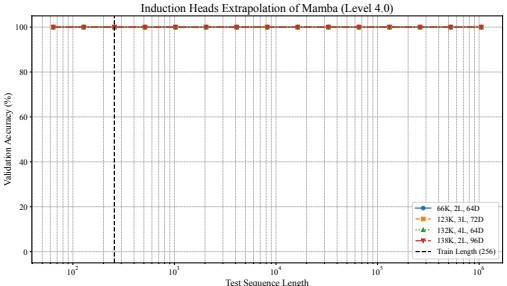 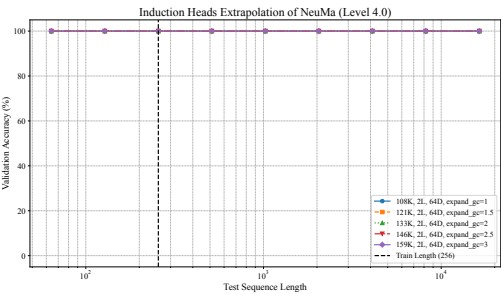

Figure 13: Sanity check validation of Mamba (left) and NeuroMamba (right) basic functionality.

**Level 4.1 (Robust Discovery)**

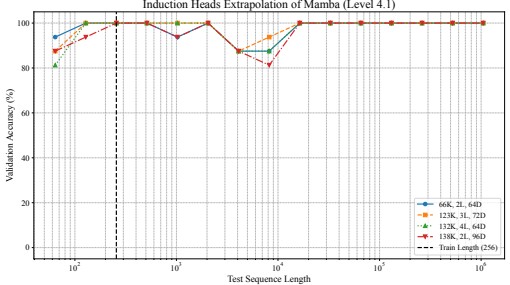 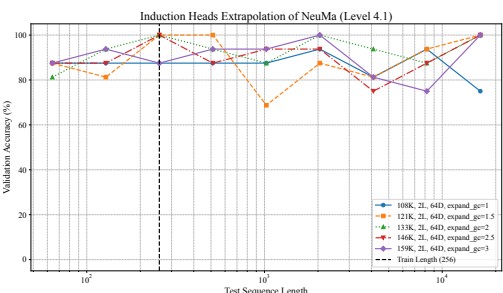

Figure 14: Robust discovery capabilities comparison between Mamba (left) and NeuroMamba (right).

**Level 4.2 (Dynamic World Modeling)**

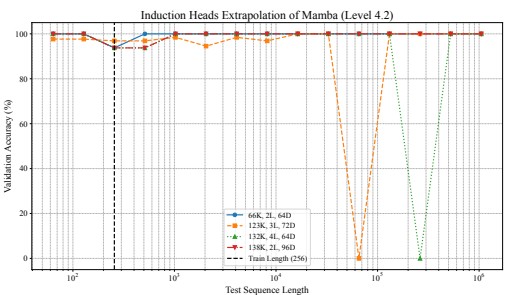 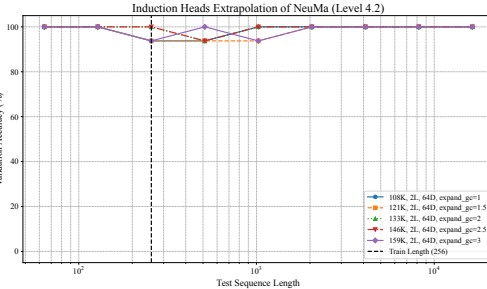

Figure 15: Dynamic world modeling performance of Mamba (left) contrasted with NeuroMamba (right).

### B.3 TWO-ALTERNATIVE CUE-DELAY-CHOICE (2ACDC) TASK

As described in Section 4.1.3, this experiment aims to validate whether NeuroMamba's circuit-level design spontaneously replicates the learning dynamics observed in the biological hippocampus. To achieve this, we created a tokenized version of the 2ACDC task and implemented a rigorous analysis protocol mirroring the original neuroscience study.

#### B.3.1 TASK IMPLEMENTATION

The 2ACDC task was translated into a next-token prediction problem. The core of the task is defined by two distinct trial templates, each represented as a fixed sequence of integer tokens.

- **Trial Type 0 ("Near Reward"):** Represented by the token sequence [1, 1, 1, 1, 1, 1, 2, 2, 2, 2, 1, 1, 1, 4, 6, 1, 1, 1, 5, 5, 1, 1, 0]. The token 2 acts as the cue, and the token 6 appears at the "near" reward location.

- **Trial Type 1 ("Far Reward"):** Represented by the token sequence [1, 1, 1, 1, 1, 1, 3, 3, 3, 3, 1, 1, 1, 4, 4, 1, 1, 1, 5, 6, 1, 1, 0]. Here, token 3 is the cue, and token 6 appears at the "far" reward location.

The complete vocabulary size for this task is 7. Each trial has a fixed length of 23 tokens. For each training epoch, a new sequence is generated by concatenating 100 trials, with the type of each trial chosen randomly. The first 50 trials of this sequence are used for training the model, while the remaining 50 trials are used for immediate, held-out evaluation of performance and internal representations within that same epoch.

#### B.3.2 MODEL AND TRAINING HYPERPARAMETERS

The NeuroMamba model architecture for this task was configured with a model dimension of 256 (d_model = 256). Other key architectural parameters were set as follows: SSM state dimension d_state = 16, 1D convolution kernel size d_conv = 4, gate controller convolution kernel size d_conv_gc = 4, and expansion factors for both the main path and the gate controller set to 2 (expand = 2, expand_gc = 2). We conducted 16 independent simulation runs to ensure the robustness of our findings, with random seeds starting from a base value of 200. Each simulation was trained for 50,000 epochs. We used the AdamW optimizer with a learning rate of 1e-4 and a Cross-Entropy loss function. Model state and performance metrics were saved every 1,000 epochs.

#### B.3.3 ANALYSIS PROTOCOL

Our analysis protocol was designed to rigorously test for the emergence of orthogonal representations, directly mirroring the methods from Sun et al. (2025). In alignment with the biological focus on the CA1 region and our model's architecture, all representational analysis was performed on the hidden states corresponding to the CA1-like module's output ($y_1$ in our implementation).

- **Success Criteria:** A simulation run was deemed successful only if it simultaneously met two strict criteria at the final training epoch: a performance threshold ($loss_{thresh} = 0.04$) and a decorrelation threshold ($decorr_{thresh} = 0.6$). The decorrelation was measured as the mean correlation of the off-diagonal blocks of the final cross-correlation matrix.

- **Temporal Sequence Analysis:** To test if NeuroMamba replicated the precise learning sequence observed in biology, we analyzed the decorrelation dynamics within specific, pre-defined temporal regions of the trial. These regions, corresponding to the analysis in Figure 6e, were defined as:

  1. **Off-diagonal:** This captures the correlation between non-identical time steps in the ambiguous "grey corridor" segments. This corresponds to the regions '[0:6]', '[10:13]', '[15:18]', and '[20:23]'.

  2. **Pre-R2:** The time steps immediately preceding the far reward, corresponding to the region '[13:15]'.

  3. **Pre-R1:** The time steps immediately preceding the near reward, corresponding to the region '[18:20]'.

For each successful run, we tracked the average correlation within these three regions across all saved epochs. The temporal learning sequence was then established by identifying the first epoch at which the average correlation for each region permanently crossed below the decorr_thresh of 0.6. The order of these "crossing points" allowed for a direct comparison with the biological findings.

### B.3.4 TEMPORAL DECORRELATION ANALYSIS OF THE MAMBA MODEL

To provide a comprehensive comparison with the NeuMa model, we conducted a temporal decorrelation analysis on the Mamba model using the same methodology. The analysis aimed to identify the epoch at which the average correlation within three key regions ("Off-diagonal," "Pre-R2," and "Pre-R1") first dropped below a biologically relevant decorrelation threshold of 0.6.

However, none of the Mamba runs that successfully passed the initial loss filter (loss ¡ 0.04) managed to achieve this level of decorrelation in any of the specified regions throughout the training process. The final correlation values remained significantly above the 0.6 threshold.

For completeness and to visualize this outcome, we present the "crossing points" analysis below using a relaxed, non-stringent threshold of 1.0 (Figure16). By definition, a correlation coefficient is always less than or equal to 1. Therefore, this condition is trivially met at the very beginning of training (Epoch 0) for any non-perfect correlation. The resulting plot graphically confirms that the Mamba model failed to undergo a dynamic, sequential decorrelation process. Instead of showing a learned temporal sequence, all crossing points converge at Epoch 0, highlighting the model's inability to sufficiently orthogonalize its representations for this task under the stringent criteria met by NeuMa.

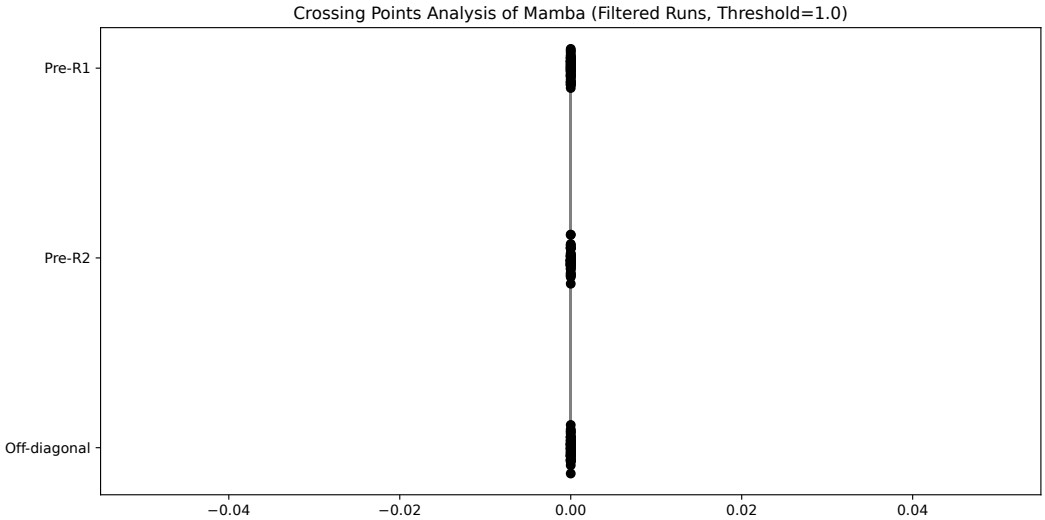

Figure 16: **Mamba model's crossing points analysis with a relaxed threshold.** The plot shows the epoch at which the correlation in key regions first dropped below a threshold of 1.0. Since this condition is met at initialization (Epoch 0), all points are at the origin. This illustrates the model's failure to achieve the required decorrelation ($< 0.6$) observed in both biological data and the NeuMa model.

# C ABLATION

Figure 17: **NeuroMamba Ablation Architectures.** Diagrams illustrating the specific architectural modifications used in the ablation studies. **(Left)** DG-like Input Ablation (ablate_gc=True): In this configuration, the modulatory DG-like input pathway is removed. **(Right)** CA3-like Output Ablation (ablate_y2=True): This configuration retains the full three-branch input structure, including the DG-like pathway. The ablation is applied to the output, where the direct CA3-like pathway is removed.

To isolate and understand the functional contributions of the key architectural components inspired by the hippocampal trisynaptic circuit—specifically, the Dentate Gyrus (DG)-like input and the CA3-like direct output pathway—we performed a series of ablation experiments, as shown in figure17. For this purpose, we implemented a modified model, NeuroMamba_ab, which allows for the selective deactivation of specific pathways through boolean flags during initialization. Crucially, for each ablation condition, the data generation, training hyperparameters, and evaluation protocols remained identical to those described in Appendix A for the Selective Copying, Induction Heads, and 2ACDC tasks. The only change was the use of the NeuroMamba_ab model with specific flags activated.

The NeuroMamba_ab class was designed to accept two primary boolean flags, ablate_gc and ablate_y2, in its constructor. These flags control the ablation of the two targeted pathways by modifying the model's architecture at initialization.

- **Ablating DG-like Mossy Fiber Input:** This ablation was implemented by setting the flag ablate_gc = True. This action triggers a specific logic within the model's constructor: the weights and biases of the mf_proj linear layer, which projects the gated controller (gc) output to modulate the SSM dynamics, are permanently set to zero. Furthermore, their gradients are disabled (requires_grad = False) to prevent them from being updated during training. This effectively severs the influence of the DG-like pathway on the core recurrent state dynamics of the CA3-like module.

- **Ablating CA3-like Direct Output:** This was achieved by setting the ablate_y2 = True flag. When this flag is active, the weights and biases of the out_cathree_proj linear layer, responsible for projecting the CA3-like state output ($y_2$) to the final model output, are zeroed out and frozen. This effectively removes this pathway's contribution to the final prediction, leaving only the CA1-like integrated output ($y_1$) to inform the model's decision.

- **Ablating Both Pathways:** The third condition, which simulates a model lacking both the DG-like input and the CA3-like direct output, was implemented by setting both ablate_gc = True and ablate_y2 = True during model initialization.

This approach ensures that the ablations are clean and complete, allowing for a precise evaluation of each component's contribution to the model's overall performance and learning dynamics across the three benchmark tasks.

## C.1 ABLATION STUDY ON THE 2ACDC TASK

As noted in the main text, a key finding of our ablation studies was the universal failure of all ablated models on the 2ACDC task. None of the variants could consistently meet the dual-threshold criteria (performance and decorrelation) required for a successful run. This failure is a significant result in itself, providing causal evidence that the complete, integrated circuit is a prerequisite for achieving biological fidelity.

Consequently, a temporal sequence analysis of their learning process, akin to that performed on the full NeuMa model (Figure **??**), would be meaningless. However, to offer a complete characterization of their behavior, we visualize the final representational state of these models in Figure 18 by plotting their average cross-correlation matrices at the final training epoch.



Figure 18: **Final Representational State of Ablated Models on the 2ACDC Task.** Average cross-correlation matrices for the three ablated model configurations. In stark contrast to the full model (Figure 6c), none of the ablated variants achieve deep, widespread representational orthogonalization. The persistent high positive (warm, red/orange colors) and negative (cool, blue colors) correlations across the matrices visually confirms their failure to form a functional "orthogonalized state machine," underscoring the synergistic necessity of the complete circuit.

# D  LANGUAGE MODELING

## D.1  EXTENDED ANALYSIS ON COMPUTATIONAL EFFICIENCY

Our findings from the head-to-head benchmark in Section 4.3.1 suggest a fundamental principle for future architectural design: the focus should be on the computational efficiency of the building block itself. NeuroMamba's structure-function principle, realized through efficient computational fusion, creates a highly expressive and hardware-friendly "superblock." It is this efficient "superblock" that allows us to build a network of equivalent capacity with significantly fewer layers (12 vs. 26). The observed leap in performance is therefore not merely a consequence of this shallowness, but an inevitable outcome of a superior local (block-level) design inducing a qualitative improvement in global (model-level) efficiency.

Figure 19 visually corroborates the findings from Table 2. The bar charts clearly illustrate Neuro-Mamba's superior mean performance across all key metrics. Crucially, the error bars, representing the standard deviation over five runs, highlight the remarkable stability of NeuroMamba's inference performance compared to the baseline Mamba, whose higher variance suggests less predictable behavior. This enhanced stability is a critical feature for deployment in real-world, latency-sensitive applications.

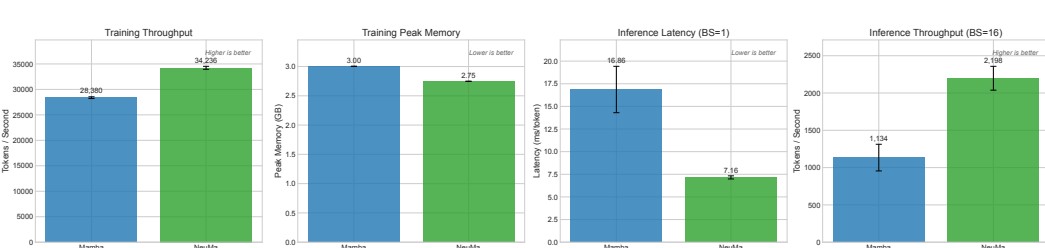

Figure 19: **Visual Comparison of Efficiency Metrics.** Bar charts visualizing the mean performance of NeuMa (blue) and Mamba (green) on key efficiency benchmarks. Error bars indicate the standard deviation across five independent runs, highlighting NeuMa's superior performance and stability, particularly in inference latency and throughput.

## D.2  NEUMA-140M PRE-TRAINING DETAILS

This section provides a comprehensive overview of the architecture, data, and training strategy used for the pre-training of the NeuMa-140M foundation model, as referenced in Section 4.3.1.

### D.2.1  MODEL ARCHITECTURE

The NeuMa-140M model is a novel language model architecture based on our innovations in selective state-space modeling.

- **Model:** NeuMa-140M

- **Parameter Scale:** Approximately 140 million total parameters.

- **Core Configuration:**
  - `hidden_size`: 768
  - `num_hidden_layers`: 12
  - `vocab_size`: 50,280 (Compatible with GPT-2/Mamba, padded for computational efficiency)

- **Normalization Layer:** RMSNorm

- **Residual Connection Precision:** `residual_in_fp32` set to `True` to maintain numerical stability during mixed-precision training.

- **Bias Terms:** Globally disabled (`use_bias=False`), following the common practice of modern large language models.

### D.2.2 TRAINING DATA

To train a versatile foundation model with multilingual and multi-domain capabilities, we employed a streaming mixture of three large-scale, high-quality public datasets.

- **Dataset Composition:**
    - **Cerebras/SlimPajama-627B:** A large-scale corpus of cleaned and deduplicated English text for general knowledge.
    - **bigcode/the-stack-dedup:** A massive, deduplicated dataset of open-source code across many programming languages. We utilized the Python subset.
    - **Skywork/SkyPile-150B:** A large-scale Chinese web text corpus.
- **Data Mixing Strategy:** We used the `interleave_datasets` function from the Hugging Face `datasets` library to dynamically mix the above datasets in a streaming fashion (`streaming=True`).
- **Sampling Probabilities:** To balance general knowledge, coding ability, and Chinese proficiency, we set the following sampling ratios:
    - SlimPajama (General English): 75%
    - The Stack (Code): 15%
    - SkyPile (Chinese): 10%

### D.2.3 DATA PREPROCESSING

- **Tokenizer:** We adopted the publicly available tokenizer from "state-spaces/mamba-130m-hf", which is a BPE tokenizer based on GPT-2 with a vocabulary size of 50,254.
- **Tokenization Process:**
    - An `eos_token` was appended to the end of each text sample to explicitly mark sequence boundaries.
    - Text was tokenized into integer IDs.
    - Sequences were packed into a fixed length of 256 tokens.

    This entire process was performed dynamically at training time using a streaming `map` operation, obviating the need for pre-storing the processed data.

### D.2.4 TRAINING STRATEGY & HYPERPARAMETERS

Our final training strategy was determined through multiple rounds of iterative optimization to balance convergence performance, training speed, and stability.

- **Optimizer:** AdamW
    - `adam_beta1`: 0.9
    - `adam_beta2`: 0.95
    - `weight_decay`: 0.1
- **Batch Size:**
    - `per_device_train_batch_size`: 32
    - `gradient_accumulation_steps`: 2
    - **Effective Batch Size:** 64 (32 x 2)
    - **Tokens per Batch:** 16,384 (64 x 256)
- **Learning Rate Scheduler:**
    - **Type:** `cosine_with_restarts`
    - **Peak Learning Rate:** 3e-4
    - **Warmup Steps:** 2,000
    - **Restart Cycles:** 5 (a reset every 30,000 steps)

- **Training Duration:**
  - **Total Training Steps:** 150,000
  - **Total Tokens Seen:** Approx. 2.5 Billion (150,000 x 16,384)
- **Training Robustness & Stability:**
  - **Mixed Precision:** `bf16=True` for acceleration and numerical stability.
  - **Gradient Clipping:** `max_grad_norm=1.0` to prevent gradient explosion.
  - **Numerical Health Checks:** A custom `HealthCheckCallback` was implemented to monitor for NaN/Inf values in model weights in real-time.
  - **Data Loading Fault Tolerance:** A custom `SafeIterableDataset` wrapper was used to automatically skip corrupted data chunks in the stream, preventing training interruptions.

### D.3    DOMAIN-ACCLIMATIZATION FINE-TUNING DETAILS

Following pre-training, the NeuMa-140M foundation model was subjected to a domain-acclimatization fine-tuning stage, as referenced in Section 4.3. The objective was to transform the model from a general-purpose text predictor into a conversational agent capable of understanding and responding to instructions within the scientific domain of chemistry.

#### D.3.1    TASK OBJECTIVE AND DATA FORMATTING

The primary goal of this stage was to teach the model instruction-following behavior. We utilized a proprietary dataset where each sample contained an instruction, an optional input, and a desired output. To adapt the model for a conversational role, we structured each sample into a multi-turn dialogue format:

```
Human: {instruction}\n{input}\n\nAssistant: {output}{eos_token}
```

To ensure the model learns to act as a helpful assistant that generates responses, rather than simply predicting the entire text block, we employed a crucial technique: **loss masking**. The loss was calculated exclusively on the tokens corresponding to the `Assistant`'s response. This was implemented by setting the labels for all prompt tokens (i.e., the `Human` turn and the `Assistant:` preamble) to -100, a value that is ignored by the standard cross-entropy loss function. This method effectively compels the model to learn the conditional generation of an answer given a prompt.

#### D.3.2    FINE-TUNING STRATEGY & HYPERPARAMETERS

The model was fine-tuned on a proprietary dataset of approximately 700,000 instruction-following examples in chemistry (`ChemData700K`). A 99/1 train/validation split was created from this data. Given that the development and application of this agent constitute a separate, ongoing research effort that will be detailed in a forthcoming publication, we do not release the fine-tuning script, dataset, or final model weights. Key hyperparameters are provided below for reproducibility of the methodology.

- **Base Model:** NeuMa-140M (from the final pre-training checkpoint-150000).
- **Optimizer:** AdamW
- **Batch Size:**
  - `per_device_train_batch_size`: 8
  - `gradient_accumulation_steps`: 4
  - **Effective Batch Size:** 32
- **Learning Rate:** 2e-5 (Constant)
- **Training Duration:** 3 Epochs
- **Sequence Length:** `max_length` was set to 256 tokens.
- **Stability & Efficiency Enhancements:**

- ○ **Gradient Checkpointing:** Enabled (`True`) to conserve memory.
- ○ **Mixed Precision:** `bf16` or `fp16` was enabled based on hardware support.
- ○ **Gradient Clipping:** `max_grad_norm=1.0`.
- **Evaluation Strategy:**
  - ○ Evaluation was performed every 1,000 steps (`eval_steps=1000`).
  - ○ The best model checkpoint was saved and loaded at the end of training (`load_best_model_at_end=True`).

### D.4 SECOND-STAGE FINE-TUNING: AGENT FOR SCIENTIFIC DISCOVERY

The final stage of our methodology involved a second round of fine-tuning to transform the domain-acclimatized chemistry model into a specialized agent for scientific discovery in piezoelectric catalysis, as referenced in Section 4.3. Given that the development and application of this agent constitute a separate, ongoing research effort that will be detailed in a forthcoming publication, we do not release the dataset, source code, or final model weights. Instead, this section outlines the methodological principles that guided its creation.

The core of this stage was the curation of a highly specialized, multi-source knowledge base designed to mirror the information landscape available to a human researcher. Our data strategy rested on three pillars:

- **Empirical Grounding:** The agent was grounded in reality through the integration of proprietary, unpublished data from our own laboratory experiments. This provided the model with direct, empirical evidence of synthesis-performance relationships that are not yet available in public literature.

- **Scholarly Foundation:** We systematically extracted and structured knowledge from decades of peer-reviewed scientific literature on catalysis, materials science, and solid-state chemistry. This endowed the agent with a deep understanding of the established theoretical and experimental consensus in the field.

- **AI-driven Hypothesis Generation:** To broaden the agent's exploratory capabilities, we employed knowledge distillation from larger, proprietary models. It is crucial to note that this was not used to directly dictate answers. Instead, knowledge distillation was used to bootstrap the agent with a broad set of plausible chemical reaction pathways and novel material hypotheses. These AI-generated hypotheses were then rigorously filtered, cross-referenced with scholarly knowledge, and validated against our proprietary experimental data, ensuring the final agent's knowledge is firmly grounded in empirical evidence.

This three-pillar approach created a synergistic information ecosystem, enabling the human-AI interactive loop described in the main text that ultimately led to the discovery of the novel catalytic process.

