# OpenReview forum: "NeuMa, Born to Work"
_ICLR.cc/2026/Conference — Submitted to ICLR 2026_

### Official Review · Reviewer_MqWf · 2025-10-29

**Soundness:** 2
**Presentation:** 2
**Contribution:** 2
**Rating:** 2
**Confidence:** 4

**Summary:**

This paper introduces NeuroMamba (NeuMa), a biologically inspired state-space model designed as a circuit-level analog of the hippocampal DG–CA3–CA1 circuits. The authors argue that existing selective SSMs (e.g., Mamba) implicitly approximate this structure but lack key functional pathways (dentate input and CA3 direct output). NeuMa explicitly implements these modules to achieve both engineering efficiency and biological interpretability. The paper provides (1) synthetic benchmark results showing improved learning stability, (2) neuroscientific validation showing that NeuMa reproduces hippocampal decorrelation dynamics, and (3) a “real-world validation” stage involving fine-tuning a pretrained NeuMa model (NeuMa-Chem) for CO_2 reduction catalysis.

**Strengths:**

- Conceptual novelty\
The paper presents an interesting perspective by interpreting state-space models through the lens of hippocampal circuitry. The proposed DG–CA3–CA1 modular structure offers a biologically inspired inductive bias that could improve interpretability and potentially influence model dynamics.

- Empirical evaluation on synthetic tasks\
On algorithmic benchmarks such as Selective Copying and Induction Heads, NeuMa achieves comparable or somewhat better convergence behavior than Mamba. The results suggest that the proposed structure can enhance stability in some settings.

- Neuroscientific analysis\
The experiments simulating hippocampal learning dynamics show qualitative similarities between NeuMa’s internal states and observed CA1 decorrelation patterns.

- Ablation study\
The ablations are generally helpful for understanding the role of each sub-module. The comparisons support the claim that DG, CA3-Out, and CA1 gating each contribute to model stability and performance, although the evidence is mainly qualitative.

**Weaknesses:**

- Ambiguous contribution of each biological module\
The paper introduces DG, CA3, and CA1 as functionally distinct components but does not provide a systematic analysis or theoretical justification of their respective computational roles. The ablation study, while informative, does not clearly demonstrate how each module contributes differently to learning dynamics or model performance. As a result, the improvement over Mamba appears somewhat ad hoc and difficult to attribute to specific mechanisms.

- Lack of quantitative rigor in the neuroscience comparison\
The reported resemblance between NeuMa’s internal dynamics and hippocampal recordings is largely qualitative. The choice of thresholds or correlation metrics used to claim “orthogonalized dynamics” is not well justified and could appear tuned to fit the narrative. A more quantitative or statistically grounded comparison would be needed to establish genuine biological correspondence.

- Unclear relevance of the “scientific discovery” example to NeuMa’s claimed capabilities\
The real-world validation section describes a CO₂ catalysis result with significant yield improvement, but it is not explained how this outcome specifically demonstrates NeuMa’s architectural advantages. It remains uncertain whether other architectures such as Transformer or standard Mamba could achieve similar predictive results under comparable training conditions. As currently written, this section feels detached from the main contribution and does not convincingly support NeuMa’s unique capabilities.

- Empirical validation remains speculative\
It is not clearly stated whether the catalytic results came from the authors’ own laboratory experiments, from simulation, or from reanalysis of prior literature. Without transparent methodology and reproducible data, the “real-world validation” claim lacks credibility.

**Questions:**

Please see the weaknesses.

---

### Official Review · Reviewer_qnSj · 2025-10-29

**Soundness:** 1
**Presentation:** 2
**Contribution:** 2
**Rating:** 2
**Confidence:** 3

**Summary:**

This work proposes an extension of the Mamba State Space architecture that is inspired by contemporary neuroscientific models of the hippocampus. The architecture introduces additional pathways that feed into an adapted SSM module. The experiments evaluate the architecture on multiple benchmark tasks in comparison to Mamba and include an ablation study into the role of the individual modules.

**Strengths:**

Establishing potential architectural links between neural circuitry and state-space models is thought-provoking and may lead to new insights for the design of next-generation architectures. The work is well motivated and makes interesting connections to the neuroscientific literature
Despite the more complex architecture, Table 2 demonstrates GPU-based training with comparable efficiency to Mamba.

**Weaknesses:**

The manuscript is using a model of the hippocampus as "blueprint" for the proposed machine learning architecture but it does not clearly separate between sources of inspiration, hypothesis, and empirical validation of the resulting approach.

Section 3 makes it difficult to understand what is established Mamba practice and what is novel. The mix of neuroscientific language (e.g. "mossy fiber") and machine learning terminology (e.g. "projection") makes it hard to discern what is description and what is motivation or inspiration.

Leaving aside the fact that the evoked contemporary theories of hippocampal computation are far from settled, even if we knew exactly how the hippocampus worked, it would not follow that the same architecture would be a good idea to implement on a modern GPU. It is indeed unlikely that the von Neumann cache hierachy illustrated in Figure 3 suits hippocampal computation (and vice versa). In fact, section 3.3 notes that the models recurrent relations are difficult to parallize and require custom kernels. This means that the performance gains of a proposed architecture must justify the downside of difficult parallelization. However, the presented experimental results, while promising, do not provide a rigorous comparison with prior work. The only baseline in Mamba but there is no comparison with alternative hybrid approaches like Jamba (Lenz+2025) and their respective baselines.

Moreover, the ablation study does not shed much light on the function of the circuit. Notably, removing the DG pathway slightly improved performance but it is not clear why. The offered hypothesis that the module is not required for simple signal filtering is one possibility but further experimentation would be required to firmly establish when including the module is a good idea.

Finally, Section 4.3.3. teases that a "success of this approach has opened a new research avenue" but leaves the  "specifics of which [to be] detailed in a forthcoming publication". Without further evidence, it is not possible to validate this claim and I recommend removing it until it can be fully explained in the planned publication.

**Additional comments**

- The work cites  Mamba as Gu & Dao, 2023 but it was published in 2024. Please ensure that you cite the published version over preprints.
- Not to get too hung up on words but to "propose a paradigm shift" (in the Kuhn sense?) might be overstating the case for what amounts to an architectural innovation of an existing deep learning model.
- I recommend to consider changing the title to be more descriptive of the work. It is currently impossible to guess from the title what the work is about.

**Questions:**

The introduction states that the "trial-and-error process [of AI design], while powerful, frequently lacks a foundation in first principles and leaves interpretability a persistent challenge". However, is that not how evolutionary search processes work? I am not sure what "first principles" went into the design of the hippocampus.

---

### Official Review · Reviewer_Sj3q · 2025-10-30

**Soundness:** 2
**Presentation:** 3
**Contribution:** 2
**Rating:** 2
**Confidence:** 3

**Summary:**

The authors present a development to the Mamba model, inspired by the neural architecture of the hippocampus. The CA1 and CA3 pathways are equated to the SSM block - CA3 is the recurrent block which utilises input-dependent A (rec) and B (inp) matrices & maintains a hidden state, while CA1 essentially utilises matrix C to multiplicatively gate the CA3 output. DG is an entirely new pathway that does not exist in current Mamba formulations, and projects an additive input-dependent contribution to the SSM block.

The model performs better on toy problems involving pattern recognition when under constraints such as noise and distractors. The model also reproduces a hippocampal finding in which responses are decorrelated when taking two paths with different futures (as in key splitter cell literature). Finally, the authors provide preliminary evidence of the model assisting with scientific discovery, although this seems slightly tangential to the main scope of the paper.

**Strengths:**

- The model displays impressive performance gains on toy problems over the standard Mamba architecture.
- The design itself is original, as are the parallels drawn to the hippocampal architecture.
- Improved model can also be implemented with the optimised parallel scan
- Ablation studies provide coarse evidence for the necessity of the components of the new model over baseline Mamba
- Diagrams of the model (fig 2, 3) are clear and helpful.

**Weaknesses:**

I would like to see a more comprehensive explanation for the rationale behind model design decisions, so that it is more normative. It is true that it is inspired by hippocampus, but it is not clear how principled this translation is, or what each pathway contributes in the normative sense. There are coarse explanations in 3.1, but these are descriptive and apply more to the biological hippocampus rather than the proposed model. I think explicitly comparing the equations of the proposed model to Mamba could help, both for explanation/elucidating the functions of the pathways but also for the reader who may not have the Mamba equations at their fingertips. This might reveal principled similarities and differences: for example, does mf_t (equation for mf should also be provided in text) just perform the effect of adding identity matrix I (or some rotation) to B, since mf is a projection of x? Another example is the D matrix I believe Mamba has on the output, which doesn't appear in the model.

To really argue the model is an improvement over Mamba I think it would be nice to see the performance on a benchmark rather than just toy examples. There is the CO2 application you touch on (a cool result for sure but possibly too far from the core ideas of the paper), but to convince the reader I think you need to test on one (or several) well-known image or text-based benchmark(s). In the 140M benchmark I could only see metrics relating to efficiency and not performance.

I am skeptical of the results in figure 5c and 7; it's not clear to me why performance increases above the training performance at extremely long sequence lengths. I also don't understand why performance varies so significantly, for example Mamba has 0%, 100%, 0% performance at seq len 10^4-10^5 on induction head level 3 task. Many points also seem to be exactly equivalent between models (e.g. seq len 10^4-10^5 green line is flat and also red has same value in middle) which seems strange, as if there are very limited discrete values (perhaps multiples of vocab size 1/16) that accuracy can take - surely accuracy should be essentially continuous if it is an average over many batches/timesteps? I would also like to see error bars on these plots between separate runs, given how unstable the curves are. Some of the ablation studies in figure 7 indicate better performance without the proposed pathways (instability could be learning rate related). I am also surprised that bigger models perform worse even on training data.

If you wish to argue that the model recapitulates hippocampus (you don't necessarily have to argue this, it's fine to just use HPC as design inspiration) which figure 6 alludes to, then it would be nice to see some analysis of the separate pathways - do they exhibit similar characteristics in their responses to the HPC pathways they were based on?

**Questions:**

- Description of the induction head tasks should be in the main text/figure in my opinion, since it's one of the core results and it's not clear from the current figure (5a) - the reader can only really understand by visiting the appendix
- Would be helpful to compare to mamba equations
- Equation for mf_t
- Fig 6 requires some clarification - 6b the dotted lines could be labelled; also not clear what 6b shows, is it the correlation between hippocampal activity on near and far runs?
- What does the better match in fig6 say about the algorithmic processes in NeuMa compared to Mamba? Why is Mamba incapable of using this representation?

---

### Meta-Review · Area_Chair_bEZ5 · 2025-12-19

**Summary:**

The reviewers agree that this paper proposes an original and thought-provoking extension of the Mamba state-space architecture inspired by hippocampal DG–CA3–CA1 circuitry. All reviewers acknowledge the conceptual novelty of explicitly mapping biological pathways to components of a selective SSM and find the synthetic experiments and qualitative neuroscience analyses interesting. The model demonstrates improved stability and performance over Mamba on several toy or algorithmic tasks under noise and distraction, and the architectural design is clearly illustrated.

However, the consensus is that the current submission does not yet meet the bar for acceptance at a top-tier machine learning venue. The primary concerns center on (1) insufficiently principled or normative justification of the architectural design beyond biological analogy, (2) limited and sometimes unstable empirical validation that relies heavily on toy benchmarks without convincing large-scale or standard benchmarks, and (3) lack of quantitative rigor and clarity in the neuroscience-inspired analyses and the “real-world” application. As a result, while the idea is promising, the reviewers agree that the evidence is not yet strong enough to support the paper’s claims about general superiority over Mamba or broader impact.

**Reviewer Concerns:**

1.All reviewers emphasize that the paper relies too heavily on biological analogy without sufficiently explaining why each added pathway should improve computation from a machine learning or algorithmic perspective. The mapping from hippocampal structures to SSM equations remains descriptive rather than theoretically grounded. In particular, explicit comparisons to standard Mamba equations (e.g., the role of the DG/mossy-fiber pathway relative to existing B, C, or D matrices) are missing or incomplete.

2.The experimental evaluation is dominated by toy and synthetic tasks. Reviewers consistently note the absence of convincing results on widely used text or vision benchmarks, making it difficult to assess whether the proposed architecture offers practical advantages beyond carefully constructed settings. The large-scale (140M) experiment focuses mainly on efficiency rather than task performance.

3.Several plots exhibit highly unstable behavior, non-monotonic trends, and counterintuitive phenomena (e.g., test accuracy exceeding training accuracy at long sequence lengths, discrete-looking accuracy values, larger models performing worse). These issues are not convincingly explained and raise concerns about experimental robustness, missing error bars, and potential sensitivity to hyperparameters.

4.The hippocampal decorrelation results are largely qualitative, with unclear metrics and thresholds, and do not yet constitute strong evidence of mechanistic alignment. Similarly, the CO₂ catalysis example is perceived as tangential and insufficiently justified as evidence of NeuMa’s unique advantages, with unclear methodology and comparison to alternative architectures.

5.Reviewers find it difficult to disentangle established Mamba components from novel contributions, due to mixed neuroscientific and ML terminology. Some claims (e.g., “paradigm shift,” “opening a new research avenue”) are viewed as overstated relative to the evidence provided.

**Reviewer Scores:**

Since there is no rebuttal and discussion, there maybe no score changes in the discussion.

Reviewer Sj3q:
Likely to remain at Reject (2). While the rebuttal may have clarified some design intentions, the major concerns regarding lack of benchmarks, unclear equations, and unstable experimental results were not fully resolved.

Reviewer qnSj:
Likely to remain at Reject (2). Core objections about the weak separation between inspiration, hypothesis, and validation, as well as insufficient justification for added architectural complexity and lack of comparisons to alternative models, persist.

Reviewer MqWf:
Likely to remain at Reject (2). Although conceptually sympathetic, this reviewer’s concerns about ambiguous module contributions, qualitative neuroscience validation, and speculative real-world impact remain largely unaddressed.

---

### Decision · Program_Chairs · 2026-01-26

Reject